# Bisphenol A (BPA) Affects the Enteric Nervous System in the Porcine Stomach

**DOI:** 10.3390/ani10122445

**Published:** 2020-12-20

**Authors:** Krystyna Makowska, Sławomir Gonkowski

**Affiliations:** 1Department of Clinical Diagnostics, Faculty of Veterinary Medicine, University of Warmia and Mazury in Olsztyn, Oczapowskiego 14, 10-957 Olsztyn, Poland; 2Department of Clinical Physiology, Faculty of Veterinary Medicine, University of Warmia and Mazury in Olsztyn, Oczapowskiego 13, 10-957 Olsztyn, Poland; slawomir.gonkowski@uwm.edu.pl

**Keywords:** endocrine disruptor, stomach innervation, porcine, toxin, neurotransmitters

## Abstract

**Simple Summary:**

Bisphenol A (BPA) is a well-known endocrine disruptor, widely distributed in the environment because of its use in the plastic production process all over the world. After getting into a living organism, due to its similarity to estrogen, it affects many organs and systems, including the nervous and gastrointestinal systems. The enteric nervous system (ENS) in the wall of the gastrointestinal tract is responsible for regulation of the function of stomach and intestine. The ENS, due to a vast number of nerve cells and a high independence from the central nervous system, is often called the “intestinal” or “second” brain. It should be highlighted that the influence of BPA on the ENS has not been fully investigated so far. Therefore, the present study focuses on the effect of BPA on the ENS of the porcine stomach. It should also be highlighted that the domestic pig is a great animal model for investigations of the influence of pathological factors on the human ENS. Therefore the results of this research will help to understand the effect of BPA on the ENS of the human stomach.

**Abstract:**

Bisphenol A (BPA) is widely utilized in plastic production process all over the world. Previous studies have shown that BPA, with its similarity to estrogen, may negatively affect living organisms. It is acknowledged that BPA distorts the activity of multiple internal systems, including the nervous, reproductive, urinary, and endocrine systems. BPA also affects the gastrointestinal tract and enteric nervous system (ENS), which is placed throughout the wall from the esophagus to the rectum. Contrary to the intestine, the influence of BPA on the ENS in the stomach is still little known. This study, performed using the double immunofluorescence method, has revealed that BPA affects the number of nervous structures in the porcine gastric wall immunoreactive to vesicular acetylcholine transporter (VAChT, a marker of cholinergic neurons), substance P (SP), vasoactive intestinal polypeptide (VIP), galanin (GAL) and cocaine- and amphetamine-regulated transcript peptide (CART). The character and severity of noted alterations depended on the part of the ENS, the BPA dose, and the type of neuronal substance. Administration of BPA resulted in an increase in the number of nervous structures containing SP, GAL, and/or CART, and a decrease in the number of cholinergic neurons in all parts of the gastric wall. The number of VIP-positive nervous structures increased in the enteric myenteric ganglia, along with the muscular and mucosal layers, whilst it decreased in the submucous ganglia. The exact mechanism of noted changes was not absolutely obvious, but they were probably related to the neuroprotective and adaptive processes constituting the response to the impact of BPA.

## 1. Introduction

Bisphenol A (BPA) is widely used in the production of polycarbonate plastics and epoxy resins [1]. This substance is present in a wide range of everyday objects including food and drink containers, toys, elements of home furnishings, thermal paper, and dental products [2,3]. Moreover, BPA from plastics is observed in water, soil, and groceries [4]. BPA enters living organisms mainly through the gastrointestinal (GI) tract, but also through the skin and respiratory tract [2]. Because BPA may negatively affect the endocrine system and metabolism processes, it is considered one of the most common and most active endocrine disruptors [5].

Because of its similarity to estrogens, BPA shows a high affinity for binding to estrogen receptors and therefore may cause disturbances in the activity in many internal organs and systems [2,3,5]. Earlier investigations have shown that BPA first affects the reproductive and nervous systems. In the reproductive system, exposure to BPA contributes to disturbances in the estrus cycle and puberty, as well as causing morphological changes in the uterus through hypertrophy of the mucosa and changes in nerves supplying this organ [6]. Some studies have reported connections between BPA and endometritis and cervical cancer [7,8].

In turn, within the nervous system BPA may negatively affect synaptogenesis, inhibit the development of neurites and dendrites, and change concentrations of calcium and neuronally-active substances in neuronal cells and nerve fibers [9,10,11,12,13]. Moreover, BPA causes negative changes in higher nervous functions, including learning, memory and behavior [14]. Such changes were even observed in offspring whose mothers were subjected to one dose of BPA in perinatal period [15]. Some previous studies have reported that BPA administration raises the risk of occurrence of neurodegenerative diseases, in particular Parkinson’s and Alzheimer’s diseases [16,17].

Due to the fact that the main way route to intoxication with BPA is the GI tract, the stomach and intestine are most exposed to the harmful influence of this toxin. It is known that BPA may act as an inhibitor of intestinal motility and mucin secretion by the intestinal mucosal layer [18]. This latter activity together with BPA-induced changes in the mucosal layer, which consists of stimulation of apoptosis and inhibition of cell proliferation in the mucosa, leads to the impairment of the intestinal barrier and resultant increase in intestinal permeability [19].

The most important factor, which regulates the majority of the functions of the stomach and intestine is the enteric nervous system (ENS) [20,21]. It is part of the autonomic nervous system, and is situated in the wall of the digestive tract and composed of millions of neurons with a high degree of autonomy in relation to the brain and spinal cord [20,21]. Enteric neurons in the stomach together with glial cells form two kinds of the intramural ganglia (Figure 1). One of them is muscular ganglia (MG), which are connected to each other by a compact system of nerves and create the myenteric plexus situated in the muscular layer between circular and longitudinal muscular fibers [22]. The second type of the gastric intramural ganglia are submucous ganglia (SG). They are located in the submucosal layer near the mucosa, and unlike the MG, are not interconnected with dense nerves and do not form a plexus [22]. The enteric neuronal cells show a high degree of diversity in terms of morphology, function, and neurochemical characterization [20,23]. They take part in regulation of the functions of the GI tract not only in physiology, but also in adaptive reactions in order to respond to processes occurring during systemic and/or gastrointestinal diseases, as well as under the impact of toxic substances [12,20,21,24,25]. With regard to correlation between pathological gastrointestinal symptoms and changes in the ENS, knowledge about this issue is not extensive, but it is known that enteric neurons undergo changes during diarrhea, constipation, and vomiting [26,27,28,29].

One of such substances is BPA, which is known as a factor, that even in small doses, may change the neurochemical characterization of the intestinal nervous structures [12,13,30]. Differently to the intestine, knowledge concerning the influences of BPA on the ENS in the stomach is extremely scanty and is confined to BPA-induced changes in the number of nervous structures containing neuronal isoforms of nitric oxide synthase, which is the marker of nitrergic neurons [13].

Therefore, the purpose of the present investigation was to study the impact of two different doses of BPA on the number of gastric enteric nervous structures containing a wide range of neuronal factors, including, besides acetylcholine—the key neurotransmitter in the ENS, substance P (SP), vasoactive intestinal polypeptide (VIP), galanin (GAL), and cocaine- and amphetamine-regulated transcript peptide (CART), which are known to participate in the maintenance of homeostasis under the influence of alimentary toxic substances [22,31,32].

The obtained results will allow us to better understand the processes associated with the impact of BPA on the innervation of the stomach of the domestic pig, which, according to current knowledge, is a very good model for investigations of the mechanisms taking place in the human GI tract due to the high degree of neurochemical, morphological, and functional similarities in the organization of the ENS between these both species [33]. Therefore, the results gained in the present investigation may be the introduction to getting to know the mechanisms occurring during the impact of BPA on the human stomach.

## 2. Materials and Methods

### 2.1. Experimental Animals and Administration of BPA

In the investigation, 15 immature, female pigs of Piétrain × Duroc breed (8 week age, weighing about 18 kg) were used. Pigs were placed in pens (five pigs in each) appropriate to age and species. Pigs were fed twice a day with typical feed for piglets with unlimited access to drinking water. All activities during the experiment received the approval of Local Ethics Committee in Olsztyn (Poland) (decision numbers 28/2013 of 22 May 2013 and 65/2013/DLZ of 27 Nov 2013).

The plan of the investigation was similar to that previously described by Szymanska et al. [12]. Namely, the pigs were placed into three groups (five animals in each), a control (C) group and two experimental groups (experimental I (Ex I) and experimental II (Ex II)). Before the morning feeding, animals belonging to the control group were treated orally with empty capsules. In the same manner, the pigs in Ex I received the same type of capsules filled with BPA (bisphenol A) >99%, catalogue no: 239658-250G, Sigma-Aldrich, Poznan, Poland) at a dose of 0.05 mg/kg body weight (b.w.)/day. In turn pigs belonging to Ex II were treated with BPA at a dose of 0.5 mg/kg b.w./day. The capsules were administered to all groups for 28 days. After this period, the pigs were subjected to premedication using Stresnil (Janssen, Beerse, Belgium, 75 μL/kg of b.w.) and after about 30 min euthanized using high doses of sodium thiopental (Thiopental, Sandoz, Kundl, Austria) given into the ear marginal vein).

### 2.2. Tissue Collection and Storage

Fragments of the stomach (in the shape of a square measuring about of 3 × 3 cm^2^) located in the gastric fundus 20 cm before the pylorus were taken from all pigs immediately after death. Tissues were put into 4% buffered paraformaldehyde (pH 7.4) for 1 h in room temperature (rt) and then transferred to phosphate buffer (3 days at 5 °C) and the buffer was changed every day. After this period, the fragments were transferred into 18% phosphate-buffered sucrose and stored at 5 °C. After 3 weeks the tissues were frozen at −20 °C, cut (using HM 525, Microm International, Dreieich, Germany), and positioned on microscopic slides.

### 2.3. Immunofluorescence Labelling

Collected tissue fragments underwent double immunofluorescence labeling according to the method previously characterized by Makowska et al. [22]. Firstly, the slices were dried for 1 h (rt). Then the stomach fragments were treated with “blocking” solution (10% normal goat serum, 0.1% bovine serum albumin, 0.01% NaN_3_, 0.25% Triton x-100, and 0.05% thimerosal in phosphate-buffered saline—PBS) for another 1 h (rt), which prevented nonspecific binding of antisera. The next stage of the method consisted of the incubation of tissues with a mixture of two primary antibodies obtained from various species: (1) antibodies against a panneuronal marker (protein gene product 9.5 (PGP 9.5)) and (2) antibodies against one of the other active substances, namely substance P (SP), galanin (GAL), vasoactive intestinal polypeptide (VIP), cocaine- and amphetamine-regulated transcript peptide (CART), or vesicular acetylcholine transporter (VAChT), which is used as a marker of cholinergic neurons. The characterization of these antibodies is shown in Table 1.

Incubation was performed in the humid chamber (rt) and continued throughout the night. On the following day, the fragments of the stomach were incubated with the mixture of species-specific secondary antibodies conjugated with appropriate fluorochromes (Table 1) to visualize the complexes antigen-primary antibody. This incubation continued for 1 h (rt). The next stage of labelling consisted of covering the tissues with buffered glycerol and cover slips. Between the particular stages of labelling, the slides with slices of the stomach were washed in PBS (each rinsing lasted 3 × 10 min). To eliminate the possibility of non-specific labelling, tests of the specificity of used antibodies were carried out. These tests included routine examination of specificity, such as preadsorption, omission, and replacement tests.

### 2.4. Evaluation of the Number of Enteric Neurons and Intramural Nerve Fibers

Labelled tissues were evaluated with Olympus BX51 microscope (Olympus, Tokyo, Japan) with appropriate filter sets. Determination of the percentage of enteric neuronal cells immunopositive to particular active substances studied consisted of examination of at least 500 PGP 9.5-positive cells located in each type of the enteric plexus from each pig for the occurrence of other neuronal substance studied. In this evaluation, neuronal cells immunopositive to PGP 9.5 were considered as 100% (Figure 2 and Figure 3).

In addition, the number of intramucosal and intramuscular nerves containing particular neuronal factor was evaluated by the counting of nerve fibers in the microscope observation field with an area of 0.1 mm^2^. Fibers were counted in four sections per animal (in five fields within one section). The percentage of neurons and the number of nerves were pooled and shown as a mean ± standard error of measurement (SEM). To prevent the double counting of the same neurons and nerves, the slices of the stomach included into the study were separated from each other by at least 200 µm.

### 2.5. Statistical Analysis

The statistical analysis was made up using one-way analysis of variance (ANOVA) with Bonferroni’s multiple comparison post hoc test with Statistica 12 software (StatSoft Inc., Tulsa, OK, USA). Differences were considered as highly significant at *p* ≤ 0.01 (**) and *p* ≤ 0.001 (***).

## 3. Results

In the present investigations all neuronal factors studied were observed in the enteric nerve cells and nerve fibers located in the porcine gastric wall (Table 2 and Table 3). Moreover, both studied doses of BPA altered the number of the enteric nervous structures positive for all neurotransmitters studied (Figure 2 and Figure 3). The character and severity of changes depended on the type of the enteric ganglia, the type of the substance studied, the localization of the nerve fibers, and the dose of BPA.

### 3.1. Neurochemical Characterization of the Gastric Enteric Neurons under Physiological Conditions

In the control group the highest percentage of the enteric neurons placed both in the myenteric and submucous ganglia were immunoreactive to VAChT. This substance was found in 23.11 ± 0.19% of all PGP 9.5 positive cells in the MG and 34.23 ± 0.23% of all PGP 9.5 positive cells in the SG. Neurons containing other substances studied were less numerous (Table 2 and Figure 2). In the MG, the second largest population were cells immunoreactive to CART (15.38 ± 0.20%). Neurons containing VIP and/or GAL amounted to 14.63 ± 0.18% and 13.50 ± 0.18%, respectively, and the least numerous were cells showing the presence of SP (12.16 ± 0.11%). There was a slightly different situation in the SG (Table 2 and Figure 3) where the enteric ganglia were the second-largest group (after VAChT-like immunoreactive (VAChT-LI) cells) were neurons containing VIP (17.57 ± 0.14% of all PGP 9.5-LI neurons). The percentage of nerve cells immunoreactive to other substances studied had a comparable level and achieved 15.83 ± 0.12%, 15.16 ± 0.25%, and 15.09 ± 0.23% in the case of cells containing GAL, CART, and/or SP, respectively (Table 2).

### 3.2. The Influence of BPA on the Neurochemical Characterization of the Gastric Enteric Neurons

During the present investigation it was shown that BPA causes the decrease in the number of VAChT-LI neuronal cells in both types of the gastric ganglia (Figure 2 and Figure 3). Low doses of BPA lead to a similar degree of decrease in the number of such nerve cells (by about 3 percentage points (pp)) in both types of ganglia, and the percentage of VAChT-LI cells in pigs receiving low doses of BPA amounted to 31.02 ± 0.38 and 20.19 ± 0.26 in the SG and MG, respectively (Table 2). In turn, high doses of BPA resulted in the most noticeable decrease in the number of VAChT-LI nerve cells in the SG. The population of such neurons in this type of ganglion under the impact of high doses of BPA decreased to 25.11 ± 0.22% (by about 9 pp compared to pigs of C group), whereas in the SG it was down to 17.42 ± 0.22% (by about 6 pp).

Regarding neurons positive for CART, GAL, and/or SP, both doses of BPA resulted in an increase in their number in both types of the enteric ganglia (Figure 2 and Figure 3). In the MG, the most visible changes concerned neurons containing GAL and/or CART. The number of the first of them under the low doses of BPA achieved 18.29 ± 0.13% (an increase of about 5 pp compared to control pigs), and pigs receiving high doses of BPA achieved 21.93 ± 0.16% (an increase of about 8 pp) (Table 2). In the case of CART-LI neurons these values achieved were18.25 ± 0.14% (about 3 pp) and 21.88 ± 0.15% (about 6 pp), respectively. A slightly-less visible BPA-induced fluctuation was noted in the population of SP-LI neurons, the percentage of which under low doses of BPA increased to 14.54 ± 0.14% (by about 2 pp), and under high doses to 17.78 ± 0.22% (by about 5 pp). In the SG, low doses of BPA resulted in similar changes (a rise of about 5 pp in comparison to control animals) in populations of neurons positive for SP, GAL, and/or CART, the percentage of which amounted to 20.57 ± 0.10%, 20.45 ± 0.11% and 20.49 ± 0.15%, respectively (Table 2).

In animals receiving high doses of BPA the most visible fluctuations concerned GAL-LI neurons, the percentage of which achieved 24.37 ± 0.16% (an increase of about 9 pp compared to control pigs). The percentage of CART- and/or SP-LI neurons in animals receiving high doses of BPA amounted to 23.10 ± 0.27% (an increase by about 8 pp) and 22.80 ± 0.15% (an increase by about 7 pp), respectively (Table 2). An interesting situation was noted in the case of nerve cells immunoreactive to VIP. In the MG, BPA causes an increase in the percentage of such neurons. Under low doses of BPA the percentage of VIP-LI cells reached 17.12 ± 0.26% of all PGP 9.5-positive cells (an increase of about 3 pp in comparison to control animals), and under high doses of BPA, the percentage of VIP-LI cells was 18.81 ± 0.19% (an increase by about 4 pp). In the SG the decrease in the number of VIP-LI neurons after BPA administration was noted (Table 2). In animals received law doses of BPA their percentage amounted to 14.35 ± 0.13% (a decrease of about 3 pp compared to control pigs), and under the impact of high doses of BPA this value achieved 13.38 ± 0.14% (a decrease by about 4 pp.) (Table 2).

### 3.3. The Influence of BPA on the Neurochemical Characterization of the Nerve Fibers in the Mucosal and Muscular Layers of the Porcine Stomach

The results of this investigation also showed that BPA influences on the number of gastric intramuscular and intramucosal nerve fibers containing all neuronal substances investigated. Similarly to the enteric neurons, the number of intramucosal and intramuscular nerves immunopositive to the majority of the substances studied increased under the impact of BPA (Table 3). The most visible changes in the comparison of the control animals were noted under the high doses of BPA in the population of intramuscular and intramucosal fibers immunoreactive to VIP. The average number of such fibers per observation field in the muscular layer increased from 13.27 ± 0.08 to 22.68 ± 0.11 (on average by about nine fibers per observation field), and in the case of fibers in the mucosal layer, from 7.92 ± 0.12 to 14.47 ± 0.10 (by about seven fibers). In turn, the slightest increase (by about four fibers per observation field in the muscular and mucosal layer) was observed in the number of CART-LI nerves. Contrary to other substances studied, VAChT after BPA administration was noted in the smaller number of nerve fibers. This situation was similar to that observed in the enteric neurons. The number of VAChT-positive fibers per observation field in the muscular layer decreased from 17.98 ± 0.10 to 14.49 ± 0.12 (by about three nerves per observation field) under the low doses of BPA and to 13.66 ± 0.17 (by about four nerves per observation field) in pigs receiving high a dose of BPA. In the case of nerves containing VAChT located in the mucosal layer both doses caused similar changes, namely a decreased of about two nerves per observation field (Table 3). 

## 4. Discussion

The results gained during the present investigation have indicated that even relatively-low doses of BPA administered for a short period may influence the neurochemical coding of the nervous cells and fibers located in the gastric wall. It is worth noting that until recently the lower dose used in the present investigation (0.05 mg/kg b.w./day) was considered by the European Food Safety Authority (EFSA) as a tolerable daily intake (TDI) dose of BPA [34]. In 2015 the TDI for BPA was temporarily reduced to 4μg/kg b.w./day [35], due to the fact that some changes in immune system had been noted under the dose 0.05 mg/kg b.w./day [36]. Nevertheless, the legislation of some countries still treats the dose of BPA at the level (0.05 mg/kg b.w./day) as a TDI or reference dose for BPA [34]. The results of the present studies have indicated that the decision of the EFSA regarding the decrease in the TDI for BPA was correct because BPA at a dose of 0.05 mg/kg b.w./day is not completely neutral for mammals, since it may result in alterations in the neurochemical coding of the enteric nervous structures situated in the wall of the stomach.

The stomach is the first part of the GI tract, in which the food is digested for a longer time (about 2–4 h depending on the type of food) [37]. For this time BPA contained in food may directly act on the gastric wall. This is because the first metabolic changes of BPA in food take place in the proximal small intestine, where BPA is absorbed and partly subjected to glucuronidation in the enterocytes with the participation of Uridine 5′-diphospho (UDP)-glucuronosyltransferase enzymes [38]. It should be pointed out that in the domestic pig only glucuronic acid conjugation occurs because of low sulfate conjugation capacity in this species, but in humans sulfate conjugation in addition to glucuronic acid conjugation may take place. Both BPA-glucuronide and BPA, which has not been metabolized in the enterocytes may get into the blood and liver, where further transformation of BPA takes place [38]. In accordance with the recent studies it is acknowledged that the stomach wall can be affected not only by BPA contained in food, but also by BPA which gets into the stomach with peripheral blood [39].

Contrary to the intestine, where BPA may affect the motility, secretion, intestinal barrier, and neurochemical coding of neuronal cells and nerves in the ENS [12,13,40], knowledge about the influence of BPA on the stomach is extremely scarce. It is only acknowledged that BPA causes changes in the number of nitrergic enteric nerve cells in both types of gastric intramural plexuses [13]. Alterations in the number of nervous structures immunopositive to the individual factors were noted in the present study both in the myenteric and submucous gastric intramural ganglia, as well as in the intramuscular and intramucosal layers. This fact clearly indicates that BPA in the stomach may also act (like in the intestine) both on the motility and the mucosal activity, because it is public known that the vast number of neurons in the muscular ganglia are involved in the processes connected to gastric motility, and structures placed in the submucous ganglia mainly participate in the regulation of intestinal secretion [41].

The changes noted in the present investigation concerned all neuronal factors studied, but the type and severity of these changes were related to the substances studied and the part of the ENS involved. BPA caused the decrease in the number of cholinergic neurons and intramural nerves. This observation is in line with the previous studies concerning fluctuations of neurotransmitters and/or neuromodulators in the autonomic nervous system supplying the uterus and small intestine in response to the impact of BPA [6,12,13,40]. Moreover, the recent investigations have reported that the number of cholinergic enteric nervous structures is also reduced during various pathological and toxic stimuli [13,42,43]. Such observations may suggest that synthesis of acetylcholine, which is the primary neurotransmitter in the ENS in physiological conditions is blocked in pathological states. In the case of BPA the reduction of the number of gastrointestinal neurons containing acetylcholine, the most important neuronal factor inducing the smooth muscles contraction in the GI tract [23,44], may be based on the relaxant activity of this substance. Namely, it is acknowledged that BPA may impact the intestinal muscles and lead to their relaxation and the inhibition of gastrointestinal motility [45]. Moreover, fluctuations in the number of nervous structures positive for other substances studied located in the myenteric ganglia and gastric muscular layer can also arise from the relaxant activity of BPA, because all these substances participate in the regulation of digestive tract motility. Namely, VIP is one of the key neuronal substances inhibiting gastrointestinal muscles contraction [22,31]. A similar relaxant influence on the gastric muscles is shown by GAL [46] and SP, whose the exact impact on the smooth muscular fibers clearly depends on the types of stimulated receptors [41]. The impact of CART on the gastrointestinal muscles is less known, but the previous investigations have reported that this substance also plays a role in the regulation of the smooth muscles contraction [47]. 

The second reason for the decrease in the number of cholinergic nervous structures in the gastric wall may be associated with the direct influence of BPA on the nervous system and neurotoxic activity of this substance, manifested by the impairment of development and functioning of synapses, axons, and dendrites, as well as changes in expression of neuroprotein and ion transport [9,12,14,48].

The neurotoxic activities of BPA are also the most likely reason for alterations in the number of enteric neurons positive for other neuronal factors included into the study. Contrary to cholinergic nervous structures, the number of nerve cells and nerve fibers positive for SP, GAL, and/or CART increased under the impact of BPA. These findings are consistent with previous studies concerning the small and large intestine, where the increase in the number of such nervous structures has been noted both under the impact of BPA and during various pathological processes [12,13,26,47,49]. All three above-mentioned substances are known as factors which participate in the adaptive and protective processes occurring in the nervous tissue. It is public knowledge that GAL plays trophic roles in the nervous system and protects neurons during neurodegenerative diseases like Alzheimer’s and Parkinson’s diseases [50]. In turn, SP inhibits apoptosis in neuronal cells [51], and CART protects neurons during ischemic injuries, reduces the production of inflammatory factors in nervous tissues, and protects mitochondria against oxidative reactions [47]. In view of the above facts, the changes in the number of enteric nervous structures immunopositive to SP, GAL, VIP, and/or CART noted in the present study are probably connected with adaptive and/or protective reactions occurring under the influence of BPA and aimed at the maintenance of homeostasis, which has also been confirmed by the previous studies [31,32,49].

However, changes in the number of neuronal cells and fibers noted in the present study may also be an effect of the pro-inflammatory activity of BPA. On the one hand, such activity is relatively well known. Previous investigations have indicated that BPA not only affects the gastrointestinal mucosal layer leading to exacerbation of apoptosis, inhibition of the secretion of mucin, and disruption of the intestinal barrier, but also causes an increase in the expression of pro-inflammatory cytokines. On the other hand, all neuronal substances investigated during this study may affect the immune system. Namely, SP is a key pro-inflammatory substance which plays many important functions in the activation of the immunological system leading to an increase in the synthesis and expression of tumor necrosis factor alpha and interleukins, including IL-l and IL-6 [52]. So, the increase in the number of nervous structures immunoreactive to SP may result from the induction of the inflammatory processes by BPA. In the present investigation the increase in the number of nervous structures positive for factors showing anti-inflammatory activity has also been noted. Namely, during this investigation in all parts of the gastric wall, administration of BPA resulted in the increase in the number of nervous structures positive for GAL, which influences NK cells and modulates the synthesis of IFN-γ, IL-18, and IL-12/23 [53] and to CART, which in the nervous system reduces the expression of inflammatory factors [47]. Mechanisms in which the influence of the same external pathological factor causes an increase in the number of neurons containing both anti- and pro-inflammatory substances is not clear, but similar reaction of the ENS on toxic and pathological stimuli has been also reported in the previous investigations [24,31,54]. Moreover, it is public knowledge that BPA may cause similar alterations in the autonomic nervous system supplying the other internal organs [6,55]. Interesting findings gained in the present investigation concerned the impact of BPA on VIP-positive nervous structures—namely, an increase in the number of nerve cells immunoreactive to VIP in the myenteric ganglia, as well as VIP-positive intramuscular and intramucosal nerves with a simultaneous decrease in the number of cells positive for VIP in the submucous ganglia has been noted. These observations are different from results obtained in the previous studies, in which the increase of VIP-positive enteric nervous was noted in the intestine both under the impact of BPA and other pathological states [12,13,22,24,30], which was explained by the fact that VIP is involved in neuroprotective processes through modulation of the balance between anti- and pro-inflammatory cytokines [24,31,32].

The decrease in the number of VIP-positive cells in the submucous ganglia noted in this study can be explained in two ways. Firstly, the exact (still not completely known) roles of VIP in the submucous ganglia of the porcine stomach are different from its functions in the small and large intestine, resulting in the different reactions of VIP-positive neurons to pathological factors. Such different reactions have been described in previous studies concerning the porcine stomach [56]. Moreover, the decrease in the number of submucosal neurons containing VIP under the impact of BPA observed in the present investigation may be related to increased demand for this substance, which is known as a neuroprotective and anti-inflammatory agent in synapses and nerve endings [25]. The clear increase in the number of intramucosal and intramuscular nerves containing VIP noted in the present study seems to confirm the latter thesis.

In conclusion, the results gained during the present investigation definitely demonstrate that BPA given in relatively low doses for short periods affects the neurochemical coding of the nervous structures located in the porcine gastric wall. Changes in the number of enteric neurons positive for the particular active substances are the first signs of exposure to BPA, even before clinical symptoms of intoxication. Because of the multidirectional adverse influences of BPA and various roles of neuronal-active substances studied, the establishment of exact mechanisms that form the basis of observed changes is difficult. Probably, changes in the gastric ENS noted in the present investigation are connected with the neurotoxic and pro-inflammatory potency of BPA and aimed at neuroprotective and adaptive reactions. However, further studies are needed to completely elucidation all mechanisms related to the impact of BPA on the ENS in the stomach. 

## Figures and Tables

**Figure 1 animals-10-02445-f001:**
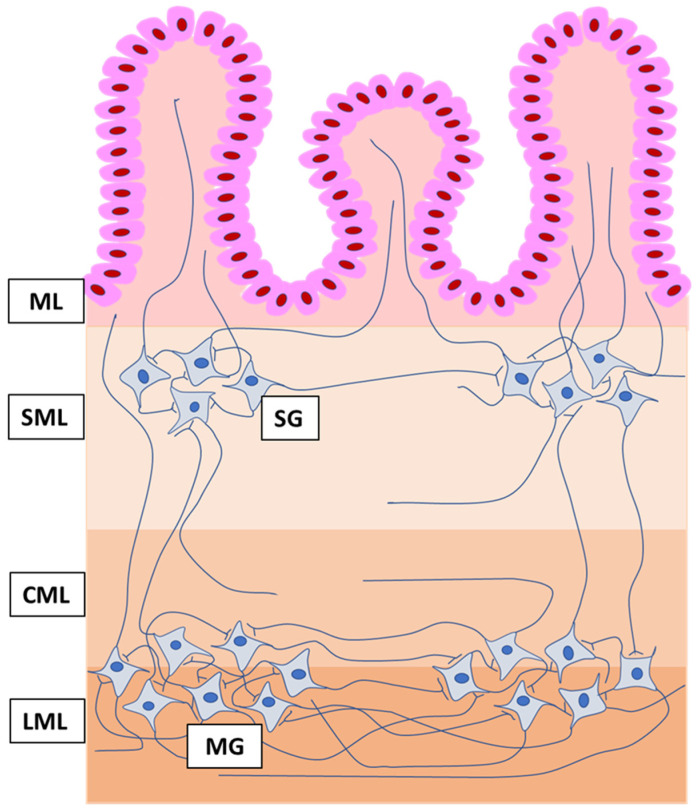
Organization of the enteric nervous system in the wall of the porcine stomach. ML, mucosal layer; SML, submucosal layer; CML, circular muscular layer; LML, longitudinal muscular layer; SG, submucous ganglia; and MG, muscular ganglia, which together with a dense network of fibers connecting ganglia, form the muscular plexus.

**Figure 2 animals-10-02445-f002:**
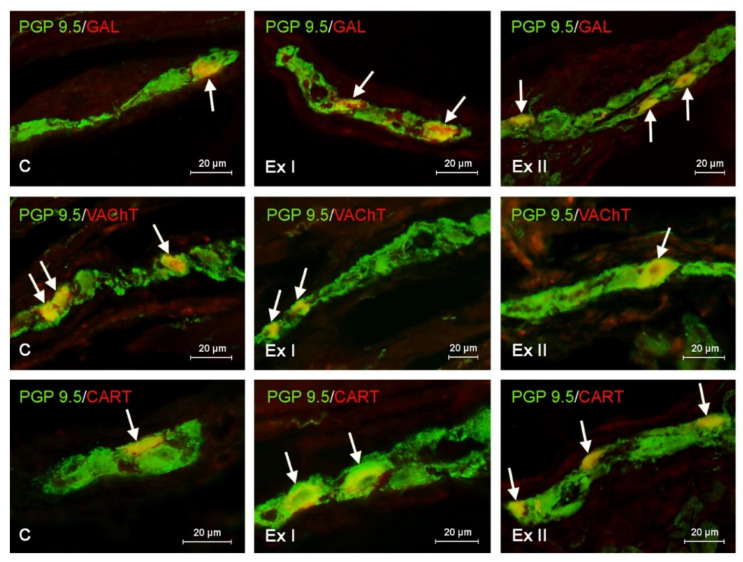
Distribution of neurons immunopositive to panneuronal marker- protein gene-product 9.5 (PGP 9.5) and other neuronal active substances—galanin (GAL), vesicular acetylcholine transporter (VAChT), or cocaine- and amphetamine-regulated transcript peptide (CART) in the myenteric plexus of porcine stomach in physiological conditions (C) and after administration of small dose (Ex I) and high dose (Ex II) of bisphenol A. The pictures are the result of the overlap of both stainings. The arrows indicate cells immunopositive for both PGP 9.5 and the other neuronal active substance studied.

**Figure 3 animals-10-02445-f003:**
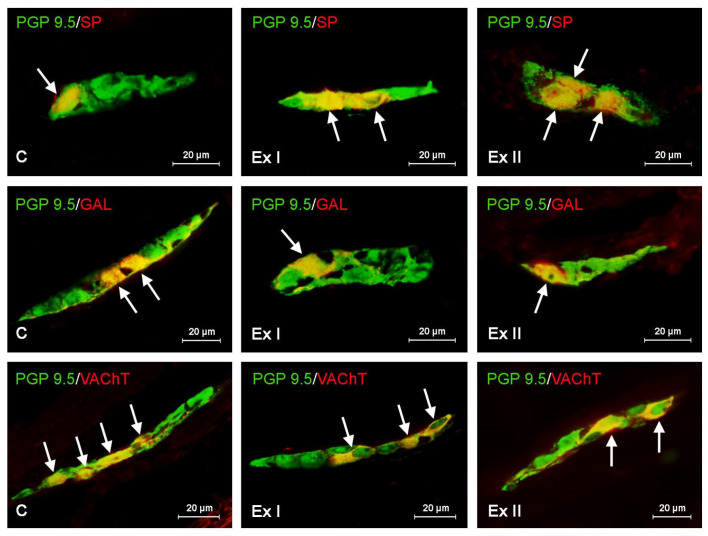
Distribution of neurons immunopositive to panneuronal marker-protein gene-product 9.5 (PGP 9.5) and other neuronal active substances—substance P (SP), galanin (GAL), or vesicular acetylcholine transporter (VAChT) in the submucous ganglia of porcine stomach in physiological conditions (C) and after administration of small dose (Ex I) and high dose (Ex II) of bisphenol A. The pictures are the result of the overlap of both stainings. The arrows indicate cells immunopositive for both PGP 9.5 and the other neuronal active substance studied.

**Table 1 animals-10-02445-t001:** Antibodies used in the present study.

**Primary Antibodies**				
**Antigen**	**Catalogue No.**	**Species**	**Working Dilution**	**Source**
CART	H-003-61	Rabbit	1:16,000	Phoenix Pharmaceuticals, Inc., Burlingame, CA, USA
GAL	AB2233	Guinea Pig	1:2000	EMD Millipore, Burlington, MA, USA
PGP9.5	7863-2004	Mouse	1:1000	Bio-Rad, Hercules, CA, USA
SP	8450-0505	Rat	1:1000	Bio-Rad
VAChT	H-V007	Rabbit	1:2000	Phoenix Pharmaceuticals
VIP	9535-0204	Mouse	1:2000	Bio-Rad
**Secondary Antibodies**				
**Reagent**			**Working Dilution**	**Source**
AF 488 donkey anti-mouse IgG (H + L)			1:1000	Thermo Fisher Scientific, Waltham, MA, USA
AF 546 goat anti-guinea pig IgG (H + L)			1:1000	Thermo Fisher Scientific
AF 546 goat anti-rabbit IgG (H + L)			1:1000	Thermo Fisher Scientific
AF 546 goat anti-rat IgG (H + L)			1:1000	Thermo Fisher Scientific

CART, cocaine- and amphetamine-regulated transcript peptide; GAL, galanin; PGP9.5, protein gene product 9.5; SP, substance P; VAChT, vesicular acetylcholine transporter; VIP, vasoactive intestinal peptide; and AF, Alexa Fluor.

**Table 2 animals-10-02445-t002:** The distribution of enteric neurons immunopositive to neuronal active factors in the porcine stomach.

**SP**			
	**C**	**Ex I**	**Ex II**
**MG**	12.16 ± 0.11	14.54 ± 0.14 *^#^	17.78 ± 0.22 *^#^
**SG**	15.09 ± 0.23	20.57 ± 0.10 *^#^	22.80 ± 0.15 *^#^
**VIP**			
	**C**	**Ex I**	**Ex II**
**MG**	14.63 ± 0.18	17.12 ± 0.26 *^#^	18.81 ± 0.19 *^#^
**SG**	17.57 ± 0.14	14.35 ± 0.13 *^#^	13.38 ± 0.14 *^#^
**GAL**			
	**C**	**Ex I**	**Ex II**
**MG**	13.50 ± 0.18	18.29 ± 0.13 *^#^	21.93 ± 0.16 *^#^
**SG**	15.83 ± 0.12	20.45 ± 0.11 *^#^	24.37 ± 0.16 *^#^
**VAChT**			
	**C**	**Ex I**	**Ex II**
**MG**	23.11 ± 0.19	20.19 ± 0.26 *^#^	17.42 ± 0.22 *^#^
**SG**	34.23 ± 0.23	31.02 ± 0.38 *^#^	25.11 ± 0.22 *^#^
**CART**			
	**C**	**Ex I**	**Ex II**
**MG**	15.38 ± 0.20	18.25 ± 0.14 *^#^	21.88 ± 0.15 *^#^
**SG**	15.16 ± 0.25	20.49 ± 0.15 *^#^	23.10 ± 0.27 *^#^

Values are shown as % (mean ± SEM) of neurons immunoreactive to substance P (SP), vasoactive intestinal polypeptide (VIP), galanin (GAL), vesicular acetylcholine transporter (VAChT, a marker of cholinergic neurons) and cocaine- and amphetamine-regulated transcript peptide (CART) in relation to all counted nerve cells in the myenteric plexus (MG) and submucous plexus (SG). Statistically-significant differences (*p* < 0.05) between control animals (C group) and animals receiving a low dose of BPA (Ex I group) as well as between C group and animals treated with a high dose of BPA (Ex II group) are marked with *. Statistically-significant differences (*p* < 0.05) between Ex I and EX II groups are marked with ^#^.

**Table 3 animals-10-02445-t003:** The distribution of intramural nerve fibers containing selected neuronal active substances in the porcine stomach.

**SP**			
	**C**	**Ex I**	**Ex II**
**ML**	8.45 ± 0.19	10.30 ± 0.09 *^#^	12.91 ± 0.09 *^#^
**CML**	9.62 ± 0.10	10.64 ± 0.09 *^#^	14.27 ± 0.09 *^#^
**VIP**			
	**C**	**Ex I**	**Ex II**
**ML**	7.92 ± 0.12	10.71 ± 0.10 *^#^	14.47 ± 0.10 *^#^
**CML**	13.27 ± 0.08	15.30 ± 0.10 *^#^	22.68 ± 0.11 *^#^
**GAL**			
	**C**	**Ex I**	**Ex II**
**ML**	6.26 ± 0.10	8.96 ± 0.10 *^#^	12.39 ± 0.14 *^#^
**CML**	9.33 ± 0.09	11.47 ± 0.15 *^#^	14.05 ± 0.09 *^#^
**VAChT**			
	**C**	**Ex I**	**Ex II**
**ML**	14.23 ± 0.10	12.96 ± 0.05 *	12.16 ± 0.06 *
**CML**	17.98 ± 0.10	14.49 ± 0.12 *	13.66 ± 0.17 *
**CART**			
	**C**	**Ex I**	**Ex II**
**ML**	11.90 ± 0.07	13.65 ± 0.09 *^#^	15.43 ± 0.07 *^#^
**CML**	17.73 ± 0.18	19.21 ± 0.05 *^#^	21.12 ± 0.09 *^#^

Values are presented as the number of nerves immunoreactive to substance P (SP), vasoactive intestinal polypeptide (VIP), galanin (GAL), vesicular acetylcholine transporter (VAChT, a marker of cholinergic neurons) and cocaine- and amphetamine-regulated transcript peptide (CART) per observation field (mean ± SEM) in the muscular layer (ML) and the mucous layer (CML). Statistically-significant differences (*p* < 0.05) between control animals (C group) and animals receiving a low dose of BPA (Ex I group) as well as between C group and animals treated with a high dose of BPA (Ex II group) are marked with *. Statistically-significant differences (*p* < 0.05) between Ex I and EX II groups are marked with ^#^.

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
