# Peer review of "Bisphenol A (BPA) Affects the Enteric Nervous System in the Porcine Stomach"

_animals, 2020, doi:10.3390/ani10122445_

Round 1
Reviewer 1 Report
L105: A comma should be added after “~used”.
L140: Tab.1 → Table 1
L141: Omit a comma after “1h”.
L143: Omit a comma after “slips”.
L144: Omit a comma after “min”.
L 151: After the sentence, it better to refer the Figure 1. Change the text accordingly. “~ as 100% (Figure 1)”. Also,
L288: Continue with the previous paragraphs. No line breaks.
L299: After the sentence, authors should refer the Table 2. “~the impact of BPA (Table 2)”.
L267~269: In the case of pigs, it is fine only glucuronic acid conjugation can be considered because of their low sulfate conjugation capacity, but in the case of humans, sulfate conjugation in addition to glucuronic acid conjugation may have to be taken into account.
L354: Change the paragraph from “The decrease in the ~”.
L367~368: What clinical manifestations (e.g., loss of appetite and vomiting) can a change in the number of enteric neurons bring about?
Author Response
The authors thank for the revision, which allows to improve the manuscript.
All editorial, linguistic and grammar suggestions of the Reviewer have been taken account
Links to tables and figures in the text have been added in accordance with Reviewer’s suggestions
Information about low sulfate conjugation capacity in the domestic pig has been added according to the suggestion of the Reviewer (lines 314-317)
Information about correlation between gastrointestinal symptoms and changes in the ENS has been added according to the suggestion of the Reviewer (lines 85-87)
Moreover, the manuscript has been significantly reworded. Among others additionally figures (Fig 1 and Fig 3) have been added, additionally table (table 1) has been added, subheadings in materials and methods and results have been added
The manuscript has been checked with regard to linguistic, grammar and editorial mistakes
Reviewer 2 Report
This paper had investigated that effects of Bisfenol A (BPA) on the enteric nervous system in the porcine stomach. The purpose of the present investigation was to study the impact of two different doses of BPA on the number of gastric enteric nervous structures, to better understand the processes associated with the impact of BPA on the innervation of the stomach of the domestic pig, and to know the mechanisms occurring during the impact of BPA on the human stomach. The experimental design seems to be OK, although I don't quite understand what these indicators mean. The writing of this paper needs further improvement. For example, there are some punctuation errors, and there are no subheadings in the Materials and methods and Results sections.
Line 72-85, Background information of enteric nervous system is very important here, and many readers don't know it. Please explain it in more detail. It is better to draw a diagram to show the relationship between these indicators measured in the Table 1 and 2 of this study.
Line 97, “processesassociated”
Line 105, “were.used” change to “were used.” How many kg of body weight for these pigs?
Line 116, “In turnpigs belonging” change to “In turn pigs belonging”
Line 104-160, Materials and methods, please add subtitle for this part.
Table 1 and Table 2, how about the statistically significant differences (p < 0.05) between Ex I group and Ex II group? Please use ". "instead of "," for the decimal point.
Line 226-227, In this paragraph, only one sentence?
Author Response
The authors thank for the insightful review, which allows to improve the manuscript
The manuscript has been extensively reworded to better describe materials and methods and show obtained results.
Subheadings in “Material and methods” and “Results” have been added according to Reviewer’s suggestion.
The scheme of organization of the ENS in the porcine stomach has been added (Figure 1)
All suggestions of the Reviewer concerning grammar, linguistic and editorial issues have been taken into account
Moreover, to better elucidation of material and methods and results additional table (table 1) and figure (figure 3) have been added.
The weight of pigs used in the experiment has been added (line 124)
Statistically significant differences between Ex I and Ex II groups has been added in tables according to the Reviewer’s suggestion
The whole manuscript has been checked with regard to linguistic, grammar and editorial mistakes.
The authors hope that improvements will allow to public the manuscript in journal “Animals”
Reviewer 3 Report
The study had as purpose to study the impact of two different doses of BPA on the number of gastric enteric nervous structures containing a wide range of neuronal factors.
The study is well described, well discussed and has interesting findings:
- The knowledge about the influence of BPA on the stomach is extremely scanty and has been raised by the authors in a proper way. BPA at a dose of 0.05 mg/kg b.w./day may result in alterations in the neurochemical coding of the enteric nervous structures situated in the wall of the stomach;
- BPA given in relatively low doses for short period affects the neurochemical coding of the nervous structures located in the porcine gastric wall;
- The study showed the impact of BPA on VIP-positive nervous structures;
- Alterations in the number of nervous structures immunopositive to the individual factors were noted both in the myenteric and submucous gastric intramural ganglia, as well in the intramuscular and intramucosal layers. This fact clearly indicates that BPA in the stomach may also act (like in the intestine) both on the motility and mucosal activity.
Although, only some minor issues are pointed below (and highlighted in yellow in the attached file) in order to improve the paper even more:
- line 2 (Bisphenol)
- line 36 - check wall..
- line 41 - change the four keywords, because they are already in the title and can be easily
found. Try new ones that highlights your research.
- line 54 - delete the last "and", because is repeating again
- line 57 - if you say some studies, I am expecting more than one cited [7]
pg 8 - lines 341-342
- line 74 - delete "showed", it is useless
- line 77 - situated in
- line 97 - separate the words (processes associated)
- line 105 - were used
- line 114 - wouldn´t it be better > 99% ?
- line 128 - replace (were undergone the double) by "underwent double"
- line 149 - examination of at least
- line 224: separate (to14.35)
- lines 238 and 242 - decreased
Table 1 and 2: replace "," by "." in the values throughout the Table. For instance: 12.16 instead
of 12,16 and so on.
line 279: as well as in
- line 296: relaxant activity
- line 301: substances
- lines 332-333: "an important function" I suppose, or many important functions
- lines 341-342 - Rewrite the sentence (something is missing here):
Mechanisms, in which under the same external pathological factor the increase (? showed increase) in the amount of neurons
REFERENCES: There are so many self citations [11], [21], [22], [23], [26], [42], and [44]. Please try to reduce this amount to the more convenient and necessary references to support the present study

Author Response
The authors thank for positive review of the manuscript
All suggestions of the Reviewer concerning grammar, editorial and linguistic issues have been taken into account.
Keywords have been changed according to the Reviewer’s suggestion
Self-citations have been limited
To better understand of the results obtained during study additional figures (figure 1 and figure 3), as well as table (table 1) have been added.
The whole manuscript has been checked with regard to linguistic, grammar and editorial mistakes.
Round 2
Reviewer 2 Report
The author's response has reduced my concerns.